# From Phenotypic Hit to Chemical Probe: Chemical Biology Approaches to Elucidate Small Molecule Action in Complex Biological Systems

**DOI:** 10.3390/molecules25235702

**Published:** 2020-12-03

**Authors:** Quentin T. L. Pasquer, Ioannis A. Tsakoumagkos, Sascha Hoogendoorn

**Affiliations:** Department of Organic Chemistry, University of Geneva, Quai Ernest-Ansermet 30, 1211 Genève, Switzerland; Quentin.Pasquer@unige.ch (Q.T.L.P.); Ioannis.Tsakoumagkos@unige.ch (I.A.T.)

**Keywords:** phenotypic screening, target identification, mechanism of action, drug discovery, chemical probes, photo-affinity labeling, proteomics, genetic screens, resistance cloning

## Abstract

Biologically active small molecules have a central role in drug development, and as chemical probes and tool compounds to perturb and elucidate biological processes. Small molecules can be rationally designed for a given target, or a library of molecules can be screened against a target or phenotype of interest. Especially in the case of phenotypic screening approaches, a major challenge is to translate the compound-induced phenotype into a well-defined cellular target and mode of action of the hit compound. There is no “one size fits all” approach, and recent years have seen an increase in available target deconvolution strategies, rooted in organic chemistry, proteomics, and genetics. This review provides an overview of advances in target identification and mechanism of action studies, describes the strengths and weaknesses of the different approaches, and illustrates the need for chemical biologists to integrate and expand the existing tools to increase the probability of evolving screen hits to robust chemical probes.

## 1. Introduction

Small organic molecules with well-defined biological activity are at the heart of chemical biology research, as they allow temporal and dose-dependent control of their target. However, to be able to reliably and reproducibly use a bioactive molecule as a probe it is necessary to understand its mode of action (MOA) and elucidate its cellular on- and off-targets [1,2]. Advances in bio-orthogonal ligation strategies, and system-based approaches such as proteomics and genome-wide gene editing, have led to a highly interdisciplinary toolbox for chemical biologists to advance poorly defined hit compounds to high quality chemical probes [3,4,5,6]. Similarly, these tools can be employed in drug discovery research, to identify new drug targets and drugs with unprecedented mode of action, or to uncover the mode of action of poorly understood marketed drugs [7,8]. Here, we review the recent developments in strategies aimed at target deconvolution and mode of action studies of bioactive molecules, focusing on methodologies that address compound action in the context of a living cell.

## 2. Identification of Bioactive Molecules

### 2.1. Target-Based Approaches

There are two main ways to establish new chemical entities to perturb a biological process of interest, a target-based approach or a phenotypic approach [9]. In a target-based approach, there is an established protein of interest (POI), and the goal is to identify a compound that will bind and perturb this POI [10]. The POI is selected based on prior knowledge of its biology, such as an implicated or established role in disease pathology. This allows for a rational approach, where compounds can be designed in silico, using molecular modeling, followed by synthesis and biochemical evaluation of compound-target binding [11]. Alternatively, in cases where it is hard to predict the compound binding site, a library of compounds or fragments (see Section 2.3) can be screened against the purified protein target [12,13]. In some cases, the target is overexpressed in a cell system, or cells are engineered for a target-specific readout, blurring the line between target-based screening and phenotypic approaches as discussed below [14,15]. 

### 2.2. Phenotypic Screens

While target-based approaches continue to yield many successful candidates for drug discovery and chemical biology research, over recent years, phenotypic screening approaches have gained traction [16]. In a phenotypic screen, there is a measurable effect on cells/organisms when a biological process is active. This effect, depending on the process of interest, ranges from growth/cell viability to the expression of a reporter such as luciferase or a fluorescent protein [17]. The system is then subjected to a library of compounds, and the effect of the compound on the phenotype is read-out. There are several advantages to this approach. First, proteins are kept in their native environment, so compound action is not limited to binding events of single proteins, but can also affect, for example, protein–protein interactions. Second, there is no bias towards a single protein target, and therefore, phenotypic screens have the potential to yield compounds that have a thus far unrecognized target or novel mechanism of action. Third, compounds exerting their effect not through protein binding but through modulation of other classes of biomolecules (i.e., DNA, RNA, lipids, or carbohydrates) can be identified. 

For example, phenotypic screens have been very powerful to identify compounds that inhibit cellular signaling pathways, such as Hedgehog [18], Wnt [19,20], STING [14], and NFκB [21] signaling. Signaling pathways are complex cellular protein networks essential for communication between cells, establishing, e.g., proper embryonic development, tissue homeostasis, and immune response, and much of the associated biology remains to be unraveled. Dysregulation of signaling pathways lies at the basis of human developmental disorders as well as diseases later in life, such as cancer. The identification of novel compounds and their associated targets thus serves multiple purposes. First, these compounds can be used to spatiotemporally control pathway activity at different levels, allowing precise perturbation and dissection of signal transduction. Second, they can form a starting point for drug development. For example, current drugs for Hedgehog pathway-dependent cancers target the upstream activator Smoothened, but mutations in Smoothened or downstream pathway components lead to tumors refractory to the therapy [22,23,24]. Discovery of compounds with a different mode of action and/or downstream target is thus needed to overcome the resistance mechanisms. 

The main disadvantage of phenotypic screening approaches is that it is often a challenging and laborious process to determine the cellular target and MOA of a screen hit (“finding the needle in the haystack”). In this review, we discuss recent examples that showcase the advances in available methodologies to understand compound action in a complex biological background. There is no universally applicable approach, yet the increasing interdisciplinary character of this area of research, including organic chemistry, proteomics, and genetics, has created ample opportunity to combine different techniques. These target deconvolution approaches are broadly applicable as they find their use not only downstream of phenotypic screens but are also vital for target-based hits to demonstrate target engagement and on-target action in the context of a live cell or organism [25,26]. Furthermore, they can be used to dissect polypharmacology or resolve controversial targets of existing drugs [27]. These methods are at the core of advancing our understanding of small molecule action in their target’s native environment, which is necessary to create high-quality and versatile chemical probes [1].

### 2.3. In-Cell Fragment-Based Ligand Discovery

Fragment-based ligand discovery (FBLD) has the advantage of covering a large range of chemical space with limited amounts of fragments, compared to small molecules screening. However, typically, a fragment library needs to be screened at >100 μM because of poor affinities, and biophysical methods are used to determine binding to purified proteins [28,29]. Recently, the Cravatt group has implemented a chemoproteomic strategy based on FBLD that bridges the gap between target- and phenotypic-based screening approaches. For this, they developed a library of fragments equipped with a photo-active group and a functional group for bio-orthogonal labeling. This library could then be used on cells to assess proteome-wide interactions with the fragments in a native context, leveraging the advantage of phenotypic screening approaches, while being able to pull-down and analyze bound targets directly through quantitative proteomics, obviating the need for lengthy downstream target deconvolution strategies [30]. A second library using enantiomeric probe pairs proved superior in identifying true probe-protein interactions (rather than aspecific binding events) [31]. While it is still early days for this screening approach, and improvements in and extension of the fragment library will enhance its overall applicability, it shows great potential for unbiased assessment of the ligandability of the proteome, thereby enhancing the options for chemical probe development.

## 3. To Label or Not to Label?

Target deconvolution strategies can be roughly divided into two areas: label-based and label-free, which will be discussed in detail below in Section 4 and Section 5. The “label-based”-category relies heavily on synthetic organic chemistry, as it requires the introduction of various functional groups in the lead molecule. “Label-free” approaches do not require synthetic alterations of the compound, which makes these methods more accessible for research groups that do not have a synthetic infrastructure or expertise. Furthermore, synthetic alterations can greatly affect the physicochemical and pharmacological properties of the lead compound and so careful controls are needed to ascertain that the probe reliably reports on the action of the lead compound. In many cases, however, structural optimization of hit compounds is still necessary to arrive at compounds with suitable potencies for in vivo studies [8]. 

## 4. Label-Based Approaches

### 4.1. Design of Covalent Probes: Activity- and Affinity-Based Probes

Synthetic organic chemistry is an important pillar of chemical biology, as it allows the precise modification of lead molecules to incorporate functionalities of interest. When using a label-based approach for target identification, the basic underlying principle is that the affinity of the compound for its target allows for the physical isolation of said target, using the compound as bait (Figure 1). Historical approaches included the immobilization of hit compounds on solid support, followed by incubation with a cellular lysate and identification of bound proteins [32]. While sometimes successful, this method cannot be used to study target engagement in a living system, nor will it be able to capture weak interactions (as low affinity binders are easily washed away). 

To overcome these limitations, the attention has shifted to methods that use a covalent interaction between the compound and its target. This covalent interaction is either established by the reaction of an intrinsic electrophilic group in the parent molecule (for example: epoxides [33,34], carbamates [35], Michael acceptors [36]) that can react with a nucleophilic residue in the protein target, or by introduction of a photo-reactive group that crosslinks upon irradiation with UV light (Figure 1b). The first category is comprised of covalent inhibitors and activity-based probes, and the second of photo-affinity probes, as discussed in more detail below (Section 4.3). 

Regardless of the nature of the covalent bond, the general structure of activity-based probes and affinity-based probes is the same, with (1) a reactive headgroup for covalent labeling; (2) a core structure that confers the binding specificity; and (3) a reporter group for detection. In a one-step (direct) probe, all three elements are synthetically connected before the probe is added to the biological sample. However, this adds a significant amount of bulk to the initial hit compound, which can compromise cellular uptake and target binding. An alternative strategy is to not directly include the reporter group, but to employ bio-orthogonal labeling instead (Figure 1a) [37]. In this two-step (indirect) approach, the probe is modified to include a small ligation handle that allows attachment of the reporter group once the probe has covalently bound its target. A variety of bio-orthogonal labeling strategies have been developed in recent years as extensively reviewed in references [3,38,39], including seminal Cu(I)-mediated Huisgen cycloaddition or strain-promoted cycloaddition between azides and alkynes (CuAAC and SPAAC “click” reactions [40,41]), azide/phosphine Staudinger ligation [42], and tetrazine/strained alkene inverse-electron demand Diels Alder reactions (IEDDA) [43] (Figure 1c). An additional advantage of a two-step labeling approach is that different reporter groups can be used, depending on the application.

The function of the reporter group is to isolate and/or visualize the probe-target complex. The most often used reporters are biotin, which can be pulled-down with streptavidin followed by mass spectrometry analysis or Western blot detection, and/or fluorescent dyes, which allow in-cell or in-gel visualization of the target. An additional key part of the design of the probes is the introduction of appropriate linkers to space the different functionalities [44]. 

### 4.2. Activity-Based Profiling

A well-established method to study target engagement in cells, especially for certain classes of enzymes, is activity-based protein profiling (ABPP) [37]. In ABPP, the small molecule probe contains a reactive headgroup that reacts in the enzyme’s active site by mimicking the substrate, and hijacking the enzyme’s reactivity to form a covalent compound–protein complex. The compound will only react when the enzyme is active, and not in an inactive or denatured state, hence the name “activity-based”. The other requirement of the activity-based probe is to contain a reporter group that can be used for detection and isolation of the probe-bound protein (Figure 1d) [45]. A variety of reactive warheads for specific enzyme families have been discovered over the years, leveraging the intrinsic reactivities of oxygen or sulfur nucleophiles (serine, threonine, glutamic acid, cysteine) in the active site of the enzyme. Examples include well-established groups such as fluorophosphonates (serine hydrolases) [46], vinylsulfones (proteasome) [47], epoxysuccinates (cysteine cathepsins) [48], and sugar epoxides (glycosidases) [49], as well as new developments recently reviewed by Benns et al. [50].

While ABPP has proven to be a very powerful technique to study enzymes in their native context, it is mostly applicable to enzyme inhibitors or intrinsic covalent drugs. A complementary approach is to use competitive ABPP for target elucidation, which is especially useful when searching for novel inhibitors of enzyme families for which broad spectrum activity-based probes have been developed (and in some cases are commercially available) [51]. For example, ATP- and ADP-based probes have been developed that covalently bind ATPases such as kinases, and are equipped with a desthiobiotin for pull-down and subsequent mass spectrometry analysis of bound enzymes. Li et al. employed these probes in competition with hits from a kinase-focused phenotypic screen, successfully identifying the kinase targets of the hit compounds in small cell lung cancer cell models [52]. 

### 4.3. Photo-Affinity Labeling

#### 4.3.1. Photocrosslink Groups

The majority of hits from phenotypic screens are non-covalent inhibitors. To be able to link these molecules covalently to their protein target, a synthetic, UV light-reactive group can be introduced suitable for photo-affinity labeling (PAL) [53]. The three main categories of photocrosslinkers are benzophenones, aryl azides, and diazirines (Figure 1b) [54,55]. When choosing a crosslinking group, there are several considerations: (1) wavelength and irradiation time: UV light causes photo-toxicity to cells and thus higher wavelengths are advantageous. Longer irradiation times lead to more non-specific labeling and photo-toxicity and so easily activatable groups are preferred; (2) lifetime of the reactive intermediate: short lifetimes of very reactive intermediates give rise to less non-specific protein labeling events, yet suffer from more non-productive quenching by water [56]; (3) bulkiness: the bigger the reactive group, the likelier it will interfere with protein binding. For these reasons, diazirines have gained in popularity, as they are small, very reactive, chemically stable, and irradiated at relatively high wavelengths [53,57,58]. Ultimately, the choice of photocrosslink group is dependent on the bioactive core under study and its tolerance to chemical manipulation, both from a synthetic standpoint (which crosslink group can be easily introduced?) and a biological standpoint (what modifications are allowed without losing biological activity?). It is noteworthy that each of the main crosslinking groups have a well-defined non-specific labeling profile as determined by 2D gel electrophoresis and proteomic studies, which should be taken into account when performing PAL experiments [59,60]. 

#### 4.3.2. Minimalist Linkers

A first requirement for making a PAL probe is a solid knowledge of the structure-activity relationship. Therefore, it is often necessary to synthesize a library of analogs of a hit compound to select the most optimal position(s) for synthetic modifications. As mentioned above, a reporter group or bio-orthogonal ligation handle needs to be introduced besides the crosslinking group, requiring even more synthetic alterations of the parent molecule. For this reason, a significant research effort has been directed towards the development of “minimalist” linkers that contain both a ligation handle and a photo-crosslinking group [61]. The first-generation minimalist linkers by Li et al. contained a diazirine and an alkyne in a short aliphatic chain [62]. This linker was incorporated in a selection of kinase inhibitors. The biological activity of the probes was confirmed and showed that the minimalist linker only had a minor effect on inhibitory potency. The probes were then used to profile their kinase targets in cells, using proteomics. Probe localization in cells was studied using fluorescent microscopy. Since Cu(I)-mediated alkyne-azide cycloadditions are incompatible with live cells, these localization studies were performed on fixed cells. To overcome this limitation, a second-generation linker was developed using cyclopropane as the ligation handle for IEDDA reactions. Incorporation of this linker in the BET bromodomain inhibitor JQ1 resulted in a covalent JQ1-based probe suitable for live cell fluorescent imaging using a “turn-on” tetrazine-functionalized fluorogenic reporter [63]. Recently, a complete suite of minimalist photo-crosslinkers was developed, and their performance validated again using BRD4 probes as a case study. The results clearly emphasized that the nature of the linker, even when quite small and “minimalist”, does influence the overall performance of the affinity-based probes [64]. The Woo lab implemented the alkyne-functionalized minimalist linker to develop a platform termed “small molecule interactome mapping by photo-affinity labeling (SIM-PAL)”, and used it to resolve the interactome of non-steroidal anti-inflammatory drugs, again combining PAL with downstream proteomics (Figure 1d) [27].

#### 4.3.3. PAL for Phenotypic Hit Target Identification

While the above examples mostly focused on in-cell target validation and off-target identification of molecules with known cellular target(s), photo-affinity labeling has also been successfully used for target identification of hits from phenotypic screens. From a focused screen, Chen et al. identified the pyrroloquinazoline LBL1 as having antiproliferative activity against breast cancer cell lines [65]. Subsequently, LBL1 was synthetically modified with either benzophenone or trifluoromethyldiazirine, with the latter being much more potent. Additional introduction of an alkyne for bio-orthogonal labeling yielded the PAL probe, LBL1-P, albeit with slightly reduced potency compared to the parent molecule. Photo-crosslinking experiments in cells with LBL1-P followed by pull-down and mass spectrometry experiments revealed that LBL1 is the first small molecule identified that targets nuclear lamins, opening up new avenues for the use of this compound to study lamin cell biology [66]. 

A recent elegant example of the usefulness of PAL for target deconvolution as well as binding site mapping is the study of Seneviratne et al. From an initial phenotypic screen for enhancers of astrocytic apoE secretion, via multiple counterscreens using relevant cell systems, the authors found a potent small molecule pyrrolidine apoE agonist (EC_50_: 57 nM). They converted their lead compound into a moderately potent PAL probe (EC_50_: 883 nM), including a benzophenone group for crosslinking and an alkyne for bio-orthogonal ligation. Using a variety of MS-based chemoproteomic methods (including CETSA, discussed in Section 5.1.3.) the cellular target of their lead compound was found to be LXRβ. This was a surprise to the authors, as they specifically counter-screened against LXRβ using a commercial kit, which in retrospect turned out be faulty and yielded a high percentage of false negatives. Nevertheless, they were successful in identifying the target of their compound using PAL [67]. 

### 4.4. FITGE

To overcome the challenge of non-specific labeling of PAL probes, Park et al. developed a method termed fluorescence difference in two-dimensional gel electrophoresis (FITGE) [68]. In this approach, a PAL probe based on the compound of interest is synthesized alongside an inactive (negative) counterpart. Cells are separately incubated with both probes, irradiated with UV light to induce crosslinking, and labeled with Cy5 or Cy3, for probe and negative probe samples, respectively. The samples are then mixed and resolved by two-dimensional gel electrophoresis. All non-specifically labeled proteins will be present as yellow (Cy3) spots on the gel, whereas red (Cy5) only spots are likely specific probe targets. These spots are then cut out of the gel and subjected to mass spectrometry analysis. The authors could demonstrate that this pipeline significantly reduces the amount of background in the MS analysis, thereby facilitating the target identification of small molecule hits from phenotypic screens [68,69,70]. The FITGE method has subsequently been expanded for background reduction in label-free approaches [71]. 

## 5. Label-Free Approaches

While activity- or affinity-based protein profiling have distinct advantages for target identification studies and functional follow-up, such as the ability to study the cellular localization of the probe and its target, and the mapping of the compound-binding site, the most important disadvantage is the synthetic effort required to make a probe. Moreover, the method is not applicable when SAR studies indicate that chemical alterations are not permitted, or when the hit compound is so chemically complex that derivatizations are not feasible, as for example, with natural products. For this reason, there has been significant interest in the development of platforms that allow target deconvolution studies without altering the chemical structure of the parent compound. Such “label-free” approaches can be divided in two main areas: mass spectrometry-based methods that rely on advances in proteomics (Section 5.1), and genetic methods, including recent CRISPR/Cas9-mediated gene editing (Section 5.2). 

### 5.1. Mass-Spectrometry Based Methods

From the above examples of PAL studies, it is clear that mass spectrometry is the method of choice to identify the proteins that are covalently bound by the probes, as it is a powerful, sensitive, and unbiased way to assess protein presence in a sample. It does suffer from false positives because of background contaminants. Many of the common contaminants have been combined in a database, “the CRAPome”, to guide users of affinity purification-mass spectrometry experiments towards bona fide protein hits [72]. Many label-free approaches also rely on high-resolution mass spectrometry to determine quantitative differences in proteomes of control samples and compound-treated samples. The overarching principle of the various methods is that compound binding results in protein conformational changes, which translates to differential chemical, thermal, or proteolytic stability (Figure 2) [73,74]. 

#### 5.1.1. DARTS

Compound binding alters the proteolytic stability of the target protein, a phenomenon which has been exploited by Lomenick et al. in a label-free method termed “drug affinity responsive target stability (DARTS)” [75,76]. In the original version, cell lysates were incubated with probe or DMSO control under native conditions to preserve protein folding. The lysates were then subjected to mild proteolytic conditions (limited proteolysis) and resolved on a 1D SDS-PAGE gel. Bands of different intensity between the two conditions were then cut out from the gel and analyzed by LC-MS/MS (Figure 2) [75]. In the years since its discovery, improvements have been made to enhance the resolution of the technique. Qu et al., in search of the target of nitazoxanide (NTZ) that was discovered as a Wnt signaling inhibitor in a phenotypic screen, used DARTS on NTZ- or DMSO-treated SW480-lysates. They subsequently resolved the samples on a two-dimensional gel and found 6 NTZ-enhanced spots. MS analysis and follow-up experiments identified PAD2 as the functional target of NTZ causing its inhibitory effect on Wnt signaling [77]. Coupling of DARTS to 2D electrophoresis can thus be useful to enhance the resolution and to identify proteins that are less abundant.

In a study by Dal Piaz and co-workers, DARTS was used to identify the cellular target of the natural product laurifolioside. The experiments were performed by incubating live cells or lysates of two different cell-lines with different concentrations of laurifolioside or DMSO control, and the results converged on clathrin heavy chain as the target [78]. This example shows that DARTS is not only applicable to ligand binding in lysates but can be extended to obtain information at the cellular level as well, by pre-incubating cells with the compound before lysing and applying DARTS. However, if the compound-target interaction is lost during cell lysis, this would compromise the subsequent DARTS analysis and so this might not be universally applicable. 

#### 5.1.2. SPROX

The group of Fitzgerald and co-workers developed a method to detect ligand-protein binding termed “stability of proteins from rates of oxidation (SPROX)” [79]. This method relies on the thermodynamic properties of proteins to fold and unfold, thereby temporarily exposing otherwise buried methionine residues. The propensity of the protein to unfold in the presence of increasing concentrations of denaturing agents such as guanidine hydrochloride or urea is assessed by the hydrogen peroxide-mediated oxidation of exposed methionines. If compound binding shifts the thermodynamic equilibrium of the protein, this will lead to a shift to the so-called “transition midpoint”: the concentration of denaturant needed to obtain equal amounts of Met-peptide:Met(ox) peptide, as determined by coupling SPROX to quantitative shotgun proteomics such as iTRAQ or TMT-labeling (Figure 2) [80,81]. 

Using this method, the targets of Manassantin A, a natural product with anticancer activity, were investigated in MDA-MB-231 cell lysates, yielding 28 potential interactors. Comparing iTraq-SPROX and a stable isotope labeling in cell culture strategy (SILAC-PP), the hit list converged on two hits, namely EF1a and filamin A [82,83]. The use of SILAC-PP coupled to SPROX (SILAC-SPROX) has the distinct advantages that the SILAC labeling is performed while cells are incubated with a compound, and so, even though the downstream SPROX process is performed on cell lysates, information is still gained at the cell level, rather than focusing on in-lysate ligand–protein binding [84]. A potential drawback of SPROX is that methionine is low abundant, which limits the attainable depth of the dataset. 

#### 5.1.3. CETSA

It has been long known that the thermal stability of proteins can change upon ligand binding, and this has been broadly used in thermal shift assays (TSA) to determine ligand binding to purified proteins. Proteins denature to varying extents when heated, leading to protein aggregates which can be separated from the soluble, native, protein population by centrifugation. If a compound stabilizes a protein, this will lead to a higher melting temperature. Consequently, the quantity of the target protein in the soluble fraction will be higher at a given temperature when comparing compound-treated to DMSO-control samples (Figure 2). 

In 2013, the Nordlund group made an important discovery when realizing that TSA could be extended to cells: proteins in a cellular context show essentially the same behavior upon heating as in isolation, with similar shifts in melting temperatures when engaged by a ligand [85]. Using the cellular thermal shift assay (CETSA), they were able to validate (off-)target engagement for a number of compounds with clinically relevant targets. In this pioneering study, cells were subjected to different concentrations of compound, and then heated to the same temperature, for the same amount of time, allowing the construction of isothermal dose-response curves. Alternatively, the drug concentration was kept constant and the cells were heated to different temperatures, cooled, and lysed. In this way, cellular target engagement of the anticancer drugs methotrexate and ralitrexed was confirmed. Importantly, drug transport could be measured as well using isothermal dose-response studies. In this initial study, the stability of target proteins was determined by Western blot, limiting the utility to target validation of known targets for which reliable antibodies exist [85,86]. 

In 2014, Savitski et al. greatly expanded the CETSA scope by combining cellular thermal shift assays with quantitative mass spectrometry analysis (Figure 2). They used MS-CETSA, also called thermal protein profiling (TPP), to profile on- and off-targets of several drugs and kinase inhibitors. Interestingly, TPP not only sheds light on direct targets but also gives information on downstream effectors, making it useful for MOA studies as well as target deconvolution [87,88]. A successful example of the use of MS-CETSA to unravel the target of a hit from a phenotypic screen is the study of Kitagawa et al. Compound a131 was found in a small-molecule phenotypic screen to kill transformed human BJ foreskin fibroblasts, while not being cytotoxic to normal BJ cells. Through MS-CETSA the authors identified PIP4K as the target of the inhibitor a131, and they were subsequently able to demonstrate the unconventional dual-action pharmacological profile of this compound [89]. 

The resolution of MS-CETSA is limited by a subset of problematic proteins that have unreliable melting curves. To overcome this, multiple groups have developed more stringent procedures, using multidimensional experiments that combine multiple compound concentrations, at different temperatures [88,90,91,92]. While these enhanced methods provide improved resolution and a wealth of data on the biological system under investigation, a drawback is the long instrumental time that is required to analyze all the different conditions. Ball et al. recently published a streamlined protocol named isothermal shift assay (iTSA), where they employed a single temperature, close to the average melting temperature of the proteome. They could then increase the number of replicates, yielding enhanced statistical power while still reducing instrument time, to identify known and additional targets of the kinase inhibitor harmine in lysates and cells of several cell lines [93]. Gaetani et al. developed a simplified TPP-like workflow, “proteome integral solubility alteration (PISA)”, where samples treated at different temperatures are combined before TMT-labeling and the area under the melting curve is measured, rather than the individual points of the curve [94]. For an excellent overview on CETSA, beyond the scope of this review, the reader is referred to the review of Dai et al. [73]. 

#### 5.1.4. FITExP

In 2015, the group of Zubarev proposed an interesting approach for small molecule target deconvolution and MOA studies which was based on their initial observation that the anticancer drug 5-FU target protein TYMS was unexpectedly upregulated during apoptosis. Therefore, they set out to see whether it could be more generally true that the protein levels of drug targets increase when compound-treated cells are in late apoptosis. If so, expression proteomics (comparison of protein abundance by LC-MS/MS of a variety of samples, see below) could indeed yield an unbiased approach towards target identification. The authors termed this methodology “functional identification of target expression proteomics (FITExP)”. In their proof-of-concept study using a panel of well-studied cytotoxic drugs, they could only identify targets when (1) excluding generic cell death proteins (by comparing proteins that were regulated under different drug conditions), as these are strongly regulated upon apoptosis in a compound-independent fashion, and (2) when using and comparing the data from multiple cell lines (using the assumption that the drug target would be present in all data sets, while non-specific proteins would vary) [95]. Recently, this group used FITExP to confirm the cellular target of the anticancer drug Auranofin, namely thioredoxin reductase I, as well as multiple indirect targets [96]. An obvious limitation of this methodology is that it requires the small molecule in question to be cytotoxic, as cells are treated at the LC_50_ to induce a significant amount of apoptosis. Furthermore, as examples are still limited, it remains to be seen if the observed protein target upregulation during apoptosis is a general feature, or limited to a selective group of drug targets.

#### 5.1.5. Large-Scale Proteomics

While FITExP is based on protein expression regulation during apoptosis, a study of Ruprecht et al. showed that proteomic changes are induced both by cytotoxic and non-cytotoxic compounds, which can be detected by mass spectrometry to give information on a compound’s mechanism of action. They developed a large-scale proteome-wide mass spectrometry analysis platform for MOA studies, profiling five lung cancer cell lines with over 50 drugs. Aggregation analysis over the different cell lines and the different compounds showed that one-quarter of the drugs changed the abundance of their protein target. This approach allowed target confirmation of molecular degraders such as PROTACs or molecular glues. Finally, this method yielded unexpected off-target mechanisms for the MAP2K1/2 inhibitor PD184352 and the ALK inhibitor ceritinib [97]. While such a mapping approach clearly provides a wealth of information, it might not be easily attainable for groups that are not equipped for high-throughput endeavors. 

All-in-all, mass spectrometry methods have gained a lot of traction in recent years and have been successfully applied for target deconvolution and MOA studies of small molecules. As with all high-throughput methods, challenges lie in the accessibility of the instruments (both from a time and cost perspective) and data analysis of complex and extensive data sets.

### 5.2. Genetic Approaches

Both label-based and mass spectrometry proteomic approaches are based on the physical interaction between a small molecule and a protein target, and focus on the proteome for target deconvolution. It has been long realized that genetics provides an alternative avenue to understand a compound’s action, either through precise modification of protein levels, or by inducing protein mutations. First realized in yeast as a genetically tractable organism over 20 years ago, recent advances in genetic manipulation of mammalian cells have opened up important opportunities for target identification and MOA studies through genetic screening in relevant cell types [98]. Genetic approaches can be roughly divided into two main areas, with the first centering on the identification of mutations that confer compound resistance (Figure 3a), and the second on genome-wide perturbation of gene function and the concomitant changes in sensitivity to the compound (Figure 3b). While both methods can be used to identify or confirm drug targets, the latter category often provides many additional insights in the compound’s mode of action. 

#### 5.2.1. Resistance Cloning

The “gold standard” in drug target confirmation is to identify mutations in the presumed target protein that render it insensitive to drug treatment. Conversely, different groups have sought to use this principle as a target identification method based on the concept that cells grown in the presence of a cytotoxic drug will either die or develop mutations that will make them resistant to the compound. With recent advances in deep sequencing it is now possible to then scan the transcriptome [99] or genome [100] of the cells for resistance-inducing mutations. Genes that are mutated are then hypothesized to encode the protein target. For this approach to be successful, there are two initial requirements: (1) the compound needs to be cytotoxic for resistant clones to arise, and (2) the cell line needs to be genetically unstable for mutations to occur in a reasonable timeframe.

In 2012, the Kapoor group demonstrated in a proof-of-concept study that resistance cloning in mammalian cells, coupled to transcriptome sequencing (RNA-seq), yields the known polo-like kinase 1 (PLK1) target of the small molecule BI 2536. For this, they used the cancer cell line HCT-116, which is deficient in mismatch repair and consequently prone to mutations. They generated and sequenced multiple resistant clones, and clustered the clones based on similarity. PLK1 was the only gene that was mutated in multiple groups. Of note, one of the groups did not contain PLK1 mutations, but rather developed resistance through upregulation of ABCBA1, a drug efflux transporter, which is a general and non-specific resistance mechanism [101]. In a following study, they optimized their pipeline “DrugTargetSeqR”, by counter-screening for these types of multidrug resistance mechanisms so that these clones were excluded from further analysis (Figure 3a). Furthermore, they used CRISPR/Cas9-mediated gene editing to determine which mutations were sufficient to confer drug resistance, and as independent validation of the biochemical relevance of the obtained hits [102]. 

While HCT-116 cells are a useful model cell line for resistance cloning because of their genomic instability, they may not always be the cell line of choice, depending on the compound and process that is studied. Povedana et al. used CRISPR/Cas9 to engineer mismatch repair deficiencies in Ewing sarcoma cells and small cell lung cancer cells. They found that deletion of MSH2 results in hypermutations in these normally mutationally silent cells, resulting in the formation of resistant clones in the presence of bortezomib, MLN4924, and CD437, which are all cytotoxic compounds [103]. Recently, Neggers et al. reasoned that CRISPR/Cas9-induced non-homologous end-joining repair could be a viable strategy to create a wide variety of functional mutants of essential genes through in-frame mutations. Using a tiled sgRNA library targeting 75 target genes of investigational neoplastic drugs in HAP1 and K562 cells, they generated several KPT-9274 (an anticancer agent with unknown target)-resistant clones, and subsequent deep sequencing showed that the resistant clones were enriched in NAMPT sgRNAs. Direct target engagement was confirmed by co-crystallizing the compound with NAMPT [104]. In addition to these genetic mutation strategies, an alternative method is to grow the cells in the presence of a mutagenic chemical to induce higher mutagenesis rates [105,106]. 

When there is already a hypothesis on the pathway involved in compound action, the resistance cloning methodology can be extended to non-cytotoxic compounds. Sekine et al. developed a fluorescent reporter model for the integrated stress response, and used this cell line for target deconvolution of a small molecule inhibitor towards this pathway (ISRIB). Reporter cells were chemically mutagenized, and ISRIB-resistant clones were isolated by flow cytometry, yielding clones with various mutations in the delta subunit of guanine nucleotide exchange factor eIF2B [107]. 

While there are certainly successful examples of resistance cloning yielding a compound’s direct target as discussed above, resistance could also be caused by mutations or copy number alterations in downstream components of a signaling pathway. This is illustrated by clinical examples of acquired resistance to small molecules, nature’s way of “resistance cloning”. For example, resistance mechanisms in Hedgehog pathway-driven cancers towards the Smoothened inhibitor vismodegib include compound-resistant mutations in Smoothened, but also copy number changes in downstream activators SUFU and GLI2 [108]. It is, therefore, essential to conduct follow-up studies to confirm a direct interaction between a compound and the hit protein, as well as a lack of interaction with the mutated protein. 

#### 5.2.2. Genetic Screens

The developments in next-generation sequencing and robust oligonucleotide synthesis have opened up novel ways to perform system-wide interrogation of gene function. RNA interference (RNAi) methods such as short interfering RNA (siRNA) and short hairpin RNA (shRNA) suffer from significant off-target effects, which lead to enhanced noise in RNAi screens [26]. This can be partially overcome with the use of large numbers of shRNAs targeting a single transcript, yet this increases the number of cells needed to conduct a genome-wide screen leading to experimental limitations [109,110]. 

With the discovery of CRISPR/Cas9-mediated editing, which provides a much more robust and direct way to manipulate gene function, the focus of genetic screens has largely shifted in that direction [111,112,113,114]. Not only can one study the effects of gene knockout by employing wildtype Cas9, gene knockdown or activation can be accomplished using mutant Cas9 protein, fused to a transcriptional repressor or activation domain, respectively [115,116]. In a pooled screening format, cells are infected with lentivirus such that each cell expresses a single shRNA/sgRNA, and the presence of the shRNA/sgRNA-encoding cassettes in the final cell pool is assessed by deep sequencing. If an essential gene is targeted, the knockout cells will not be viable and drop out of the population; conversely, if the knockout results in a growth advantage, those cells will be enriched in the final population. This approach can similarly be used to find genes that mediate sensitivity to applied perturbations, such as toxins, or as discussed in more detail below in Section 5.2.3., small molecules (Figure 3b) [109,111,112,113,114,117]. 

While most genetic screens are growth-based and study processes related to cell viability or cytotoxic compounds, we recently expanded the applicability of this methodology to study the Hedgehog signaling pathway, using an engineered cell line with a Hh-responsive reporter that confers antibiotic resistance to the cells upon pathway activation [118]. Others have used a flow cytometry rather than a growth-based approach, making use of a fluorescent reporter to isolate cells with enhanced or reduced Hedgehog signaling capabilities [119]. While these systems have not yet been employed in combination with small molecules, they are envisioned to be useful to study the mechanism of action of Hedgehog pathway inhibitors [120]. 

#### 5.2.3. “Chemogenomics”: Examples of Gene-Drug Interaction Screens

When genetic perturbations are combined with small molecule drugs in a chemogenetic interaction screen, the effect of a gene’s perturbation on compound action is studied. Gene perturbation can render the cells resistant to the compound (suppressor interaction), or conversely, result in hypersensitivity and enhanced compound potency (synergistic interaction) [5,117,121]. Typically, cells are treated with the compound at a sublethal dose, to ascertain that both types of interactions can be found in the final dataset, and often it is necessary to use a variety of compound doses (i.e., LD_20_, LD_30_, LD_50_) and timepoints to obtain reliable insights (Figure 3b). 

An early example of successful coupling of a phenotypic screen and downstream genetic screening for target identification is the study of Matheny et al. They identified STF-118804 as a compound with antileukemic properties. Treatment of MV411 cells, stably transduced with a high complexity, genome-wide shRNA library, with STF-118804 (4 rounds of increasing concentration) or DMSO control resulted in a marked depletion of cells containing shRNAs against nicotinamide phosphoribosyl transferase (NAMPT) [122]. 

The Bassik lab subsequently directly compared the performance of shRNA-mediated knockdown versus CRISPR/Cas9-knockout screens for the target elucidation of the antiviral drug GSK983. The data coming out of both screens were complementary, with the shRNA screen resulting in hits leading to the direct compound target and the CRISPR screen giving information on cellular mechanisms of action of the compound. A reason for this is likely the level of protein depletion that is reached by these methods: shRNAs lead to decreased protein levels, which is advantageous when studying essential genes. However, knockdown may not result in a phenotype for non-essential genes, in which case a full CRISPR-mediated knockout is necessary to observe effects [123]. 

Another NAMPT inhibitor was identified in a CRISPR/Cas9 “haplo-insufficiency (HIP)”-like approach [124]. Haploinsuffiency profiling is a well-established system in yeast which is performed in a ~50% protein background by heterozygous deletions [125]. As there is no control over CRISPR-mediated loss of alleles, compound treatment was performed at several timepoints after addition of the sgRNA library to HCT116 cells stably expressing Cas9, in the hope that editing would be incomplete at early timepoints, resulting in residual protein levels. Indeed, NAMPT was found to be the target of phenotypic hit LB-60-OF61, especially at earlier timepoints, confirming the hypothesis that some level of protein needs to be present to identify a compound’s direct target [124]. This approach was confirmed in another study, thereby showing that direct target identification through CRISPR-knockout screens is indeed possible [126]. 

An alternative strategy was employed by the Weissman lab, where they combined genome-wide CRISPR-interference and -activation screens to identify the target of the phase 3 drug rigosertib. They focused on hits that had opposite action in both screens, as in sensitizing in one but protective in the other, which were related to microtubule stability. In a next step, they created chemical-genetic profiles of a variety of microtubule destabilizing agents, rationalizing that compounds with the same target will have similar drug-gene interactions. For this, they made a focused library of sgRNAs, based on the most high-ranking hits in the rigosertib genome-wide CRISPRi screen, and compared the focused screen results of the different compounds. The profile for rigosertib clustered well with that of ABT-571, and rigorous target validation studies confirmed rigosertib binding to the colchicine binding site of tubulin—the same site as occupied by ABT-571 [127]. 

From the above examples, it is clear that genetic screens hold a lot of promise for target identification and MOA studies for small molecules. The CRISPR screening field is rapidly evolving, sgRNA libraries are continuously improving and increasingly commercially available, and new tools for data analysis are being developed [128]. The challenge lies in applying these screens to study compounds that are not cytotoxic, where finding the right dosage regimen will not be trivial. 

## 6. Conclusions and Perspectives

Recent years have seen a rapid rise in high-throughput “omics” technologies, which has opened up many novel ways of unbiased assessment of a small molecule’s action inside a living cell. Advances in bio-orthogonal chemistry and “minimalist” linkers for photo-affinity labeling have furthermore led to the creation of probes that very closely resemble the parent hit compound, and that can be used to confirm target engagement in cells, and follow the target localization in real time. More often than not, target identification studies yield hit lists, rather than a single hit, and complementary approaches can aid in hit prioritization [129,130]. Not discussed in detail here are computational approaches and signature databases, which have also greatly developed over the years and form an excellent additional resource to find targets or elucidate MOAs [131,132,133,134,135,136,137,138,139,140].

Whereas one of the promises of phenotypic screens is that one is not limited to find molecules with protein targets, the development of downstream target ID and MOA methods for non-protein targets remains challenging. Target-naïve image-based morphological profiling with multiplexed fluorescent dyes, “Cell Painting”, has recently been successfully used to elucidate the mode of action of autoquin. Autoquin sequesters Fe^2+^ in the lysosomes, ultimately resulting in cell death, a mechanism which would be difficult to resolve with the methodologies covered in this review. This and other examples illustrate the value of cell painting assays to unravel unconventional MOAs [141,142,143]. Profiling of RNA-small molecule interactions has become more accessible through pioneering work of the Disney group [144]. Furthermore, Mukherjee et al. recently developed a photo-affinity platform for RNA-small molecule interactions, termed PEARL-seq [145]. Another promising direction to bridge the gap between target- and phenotypic-based screening approaches is to introduce crosslink and ligation groups directly in the compound library that is used in cell-based screens, as in recent examples using fragment-based ligand discovery [30,31]. It is up to the chemical biology community to further take up this challenge and integrate and expand the existing toolbox. Ultimately, this will result in robust chemical probes for cell biological studies and drug discovery.

## Figures and Tables

**Figure 1 molecules-25-05702-f001:**
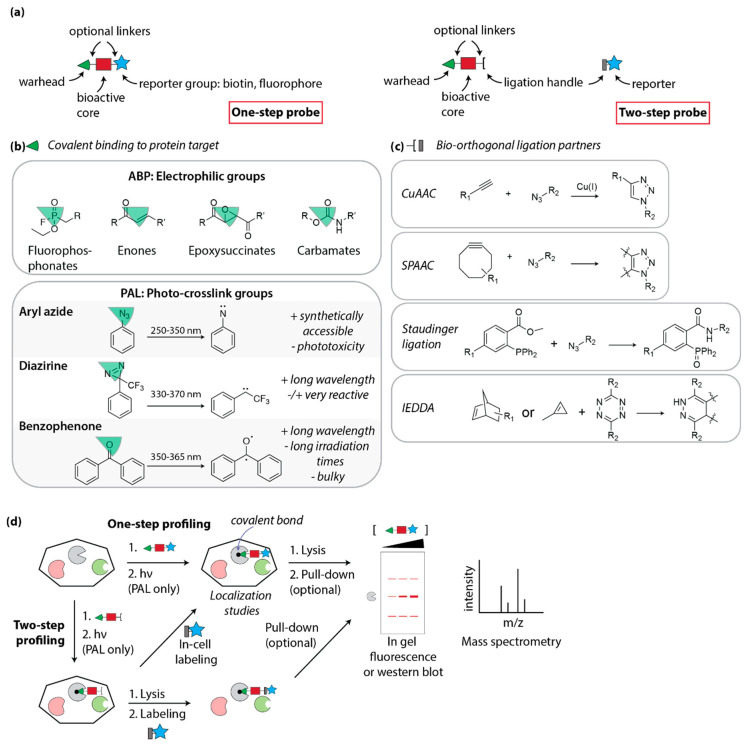
Label-based approaches. (**a**) General structures of one- and two-step probes. (**b**) Examples of warheads for activity-based probes (ABP) and photo-affinity labeling (PAL). (**c**) Examples of bio-orthogonal ligation reactions for use in two-step profiling. (**d**) Schematic representation of a cell-based profiling experiment, either through direct (one-step) or indirect (two-step) labeling.

**Figure 2 molecules-25-05702-f002:**
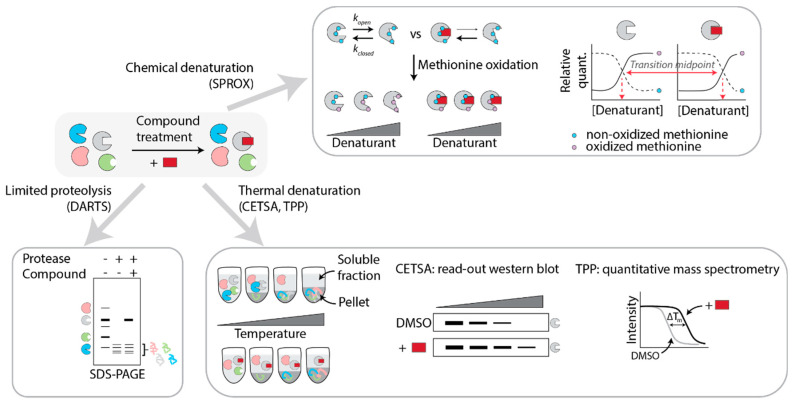
Label-free proteome-based methods for target identification and mode of action studies. Cell lysate or intact cells are treated with the compound of interest and subsequently subjected to limited proteolysis (DARTS), thermal denaturation (CETSA, TPP), or chemical denaturation (SPROX).

**Figure 3 molecules-25-05702-f003:**
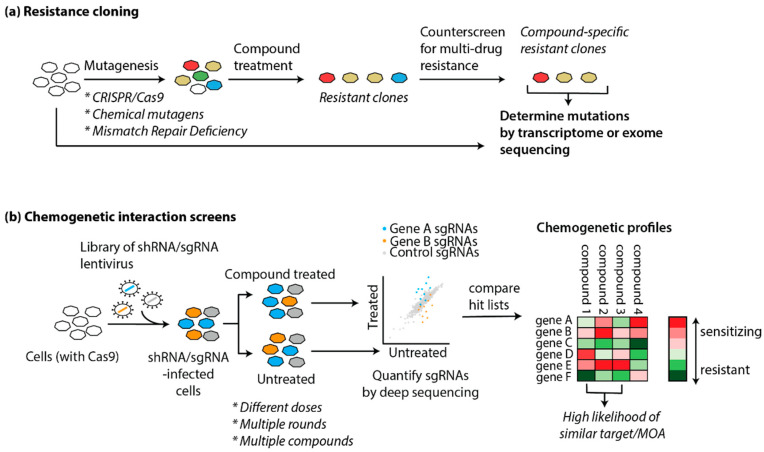
Genetic methods for target identification and mode of action studies. Schematic representations of (**a**) resistance cloning, and (**b**) chemogenetic interaction screens.

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
