# Peer review of "From Phenotypic Hit to Chemical Probe: Chemical Biology Approaches to Elucidate Small Molecule Action in Complex Biological Systems"

_molecules, 2020, doi:10.3390/molecules25235702_

Round 1

Reviewer 1 Report

In this review, Pasquer et al. survey recent advances in chemical biology approaches to identify small molecule drugs, decipher their targets and study their mode of action. They describe target-based and phenotypic approaches to discover bioactive molecules. Provide an overview of activity-based probes, protein and proteome-wide profiling as well as a range of tag-free profiling strategies. Lastly, they illustrate the latest advances in genetic and chemogenomic screens with relevant literature examples.

Overall, this is a valuable resource for chemical biologists, outlining the latest tools in small-molecule research, and the paper is well written. There are however some specific issues about the clarity of some of the techniques discussed and some general suggestions that may help create a more comprehensive description of advances in the field. Specific comments are:

  1. In section 2 the authors discuss two approaches to discover bioactive molecules. The authors make the statement that chemical entities are discovered either in a target-based approach or by a phenotypic approach. While generally true, recent advances in the field of fragment-based ligand discovery (FBLD) stretch this definition beyond the binary description. Combining the concept of photoaffinity labeling with FBLD has created a novel approach for proteome-wide screens that are analyzed by chemical proteomics. This example has opened a new avenue in proteome-wide chemical screening that does not rely on phenotypic readouts. The authors should consider adding this third arm to discovery of chemical entities and the following papers cited with relation:

Parker, C. G. et al. Ligand and target discovery by fragment-based screening in human cells. Cell 168, 527–541.e529 (2017).

Wang, Y. et al. Expedited mapping of the ligandable proteome using fully functionalized enantiomeric probe pairs. Nat. Chem. 11, pages1113–1123 (2019)

  1. In section 4.2 the authors describe the methodology of activity-based protein profiling (ABPP), the authors should add the following citation to credit the inventors:

Speers AE, Adam GC, Cravatt BF (April 2003). "Activity-based protein profiling in vivo using a copper(i)-catalyzed azide-alkyne [3 + 2] cycloaddition". Journal of the American Chemical Society125 (16): 4686–7

In the context of competitive ABPP, the authors should cite the following example:

Leung, D., Hardouin, C., Boger, D. L., and Cravatt, B. F. (2003). Discovering potent and selective reversible inhibitors of enzymes in complex proteomes. Nat. Biotechnol. 21, 687–691. 

  1. In section 5.1.4, the description the authors provide for FITExP is unclear and potentially misleading. The authors should better describe the technical method and define the concept of expression proteomics. The statement “drug targets get upregulated during apoptosis” suggest a general trend for all drugs, it should be more clearly stated that this is a specific example for the particular drug under discussion.

  1. In section 6, the authors provide concluding remarks and expand on the prospects for using advanced probes to elucidate unconventional MOAs. Several advances have been made towards expanding the scope of target-based approaches and chemical screens towards proteome-wide applications. The authors should include a short discussion of the prospects of bridging target-based approaches like covalent fragment screens with phenotypic assays to achieve proteome-wide discovery platforms.

Reviewer 2 Report

In this submitted manuscript by Pasquer et al., the authors reviewed the current available chemical biology tools and methods that are suited for target identification. The authors first glanced at how to identify phenotypic hit from high throughput screening assays. The main body of this review is to summarize how to elucidate the small molecule hits act on the biological systems. In particular, they focused on chemical proteomics methods to identify the protein target that small molecule hits bind and act on. I found this review is well organized and written. It is of sufficient significance and general interest to merit publication in this journal. However, the authors may consider the following suggestion to improve the impact of this review.

  1. In Figure 1, the authors outlined the current available bioorthogonal chemistry for covalent labeling to protein target and ligation pairs. However, I found the summary of these chemistries is limited. There are way more other reaction types. Please be detailed on these chemistries.
  2. One of the missing piece of this review is the parallel comparison of the chemistry present in Figure 1. What are the advantages and disadvantages of these reactions. What are their common protein family for targeting, eg. carbamate for hydrolases, enones for cysteine proteases etc. What are the reaction rate and lifetime for the photo-crosslinkers? What are the reaction kinetics and labeling time scale of ligation pairs? Please outline how to select these chemistries for our own studies.
